# Prevalence and associated factors of overweight/obesity among severely ill psychiatric patients in Eastern Ethiopia: A comparative cross-sectional study

**Dilnessa Fentie**[1]*, **Tariku Derese**[2]

**1** School of Medicine, College of Medicine and Health Sciences, Dire Dawa University, Dire Dawa, Ethiopia,
**2** Department of Public Health, College of Medicine and Health Sciences, Dire Dawa University, Dire Dawa, Ethiopia

* fagitaabo@gmail.com

**Data Availability Statement:** All relevant data are within the manuscript and its Supporting information files.

## Abstract

### Background

Globally, the burden of overweight and obesity is a major cardiovascular disease risk factor and is even higher among patients with psychiatric disorders compared to the general population. This is mainly due to the deleterious lifestyles characterized by physical inactivity, excessive substance use, and unhealthy diets common among patients with psychiatric disorders, as well as the negative metabolic effects of psychotropic medications. Despite these conditions being a high burden among patients with psychiatric illness, little attention is given to them during routine reviews in psychiatric clinics in most African nations, including Ethiopia. Therefore, this study aimed to estimate and compare the prevalence of and associated risk factors for overweight and obesity among patients with psychiatric illnesses.

### Methods

A comparative cross-sectional study was conducted between severely ill psychiatric patients and non-psychiatric patients in Dire Dawa, Eastern Ethiopia. The study included 192 study participants (96 psychiatric patients and 96 non-psychiatric controls). Weight and height were measured for 192 study participants. Baseline demographic and clinical characteristics of psychiatric and non-psychiatric patients were described. The data were cleaned and analyzed using the Statistical Package for Social Sciences, Version 21. The intergroup comparisons were performed using an independent sample t-test and Chi-square tests. Logistic regression analysis was used to identify the association between overweight/obesity and the associated variables.

### Results

The magnitude of overweight/obesity was significantly higher in the severely ill psychiatric groups (43.8%) than in the non-exposed controls (20.80%). The prevalence of overweight/

**Funding:** The author(s) received no specific funding for this work.

**Competing interests:** The authors have declared that no competing interests exist.

**Abbreviations:** AOR, adjusted odds ratio; BMI, body mass index; COR, crude odds ratio; WHO, World Health Organization.

obesity was highest in major depressive disorders (40%), followed by schizophrenia (32%), and bipolar disorder (28%).

## Conclusions

There was a high prevalence of obesity/overweight among psychiatric patients. Educational status, unemployment, and late stages of the disease were significant predictors of overweight/ obesity. Clinicians should be aware of the health consequences of overweight/obesity, and considering screening strategies as a part of routine psychiatric care is strongly recommended.

## Introduction

Obesity/overweight have reached an epidemic burden globally, with at least 2.8 million people dying each year as a result of being overweight or obese [1]. According to World Health Organization (WHO) European Region, an estimated 23% of women and 20% of men are obese or overweight [2]. The magnitude of overweight and obesity among adults in Africa is 27% and 8% respectively [3] and the report from the Ethiopia Demographic and Health Survey(EDHS) the burden of overweight/obesity is 8% [4]. Overweight and obesity are major risk factors for a number of chronic diseases, including diabetes, cardiovascular diseases, and cancer. Obesity is frequently accompanied by depression, and the two can trigger and influence each other. Obesity and overweight are major public health issues, as well as the leading preventable cause of death in both developed and developing countries [5].

A study has shown that the burden of obesity reveals that more than 1.9 billion adults aged 18 and older were overweight in 2014; over 600 million were obese; 39% of adults aged 18 and over were overweight, and 13% were obese [6]. Overweight and obesity are on the rise worldwide, not only in the general population but also in psychiatric patients. The major causes of overweight and obesity are not clear but are considered multi-disciplinary and related to genetic, metabolic, and psychological factors [7, 8].

The relationship between abnormal body weight and psychiatric disorders has continuous, complex relations that are still being debated. Some scholars suggest that overweight/obesity may cause common psychiatric disorders, whereas others have found that psychiatric patients are more prone to obesity [9, 10]. A community-based study of Saudi Arabian university students showed that obesity and overweight were positively associated with several mental disorders, especially mood disorders and anxiety disorders [11]. Another study done in Egypt showed that the prevalence of obesity and overweight in psychiatric patients was 66.93% (22.31% were obese, and 44.62% were overweight). The prevalence of obesity was highest in bipolar disorder (41.38%), followed by depression (37.93%), schizophrenia (10.34%), anxiety disorder (6.9%), and finally substance abuse disorder (3.45%), but the difference was not statistically significant. There was a significant correlation between the sociodemographic characteristics of patients with obesity and the distribution of psychiatric disorders [12].

In developing countries, along with economic development and income growth, the number of people who are overweight or obese is increasing. Among the reasons for the increasing obesity in the population of poor people are higher unemployment, lower education level, irregular meals, and low physical activity, which among the poor is associated with a lack of money for sports equipment [13]. In spite of such a burden of overweight/obesity in the community, there is no available scientific data in our region, Ethiopia. Thus, the purpose of this

study is to compare the prevalence of overweight/obesity in patients with severely ill psychiatric patients at an Eastern Ethiopia psychiatric center as compared to non-exposed controls. The authors hypothesized that the prevalence would be higher in patients with psychiatric disorders. This study also determined the associated factors of overweight/obesity.

## Study area and period

The study was conducted at Dire Dawa mental health center in Eastern Ethiopia from January to June 2021. The mental health center provides both inpatient and outpatient services to the community residing in the region and neighboring countries like Somalia and Djibouti.

## Study design and population

A facility-based comparative cross-sectional study was conducted between patients with psychiatric disorders and without psychiatric disorders. All clients visiting outpatient departments for treatment or medical advice during the study period were considered as the source population, whereas only selected clients attending outpatient departments for treatment or medical advice during the study period were the study population. Severely ill psychiatric patients (exposed)-established diagnoses of a common psychiatric disorder include schizophrenia, schizoaffective disorder, major depressive disorder, and bipolar disorder. The psychiatric diagnosis of the subjects was obtained from the patients' records diagnosed by psychiatrist specialist and relied on the Diagnostic and Statistical Manual of Mental Disorders(DSM-5) [14]. Non-psychiatric study groups (non-exposed group/controls) were age and sex-matched individuals, who had no lifetime diagnosis or treatment for a mental illness and who attended outpatient departments for general medical or surgical treatment other than psychiatric illnesses with in the same health institution.

## Selection criteria

All study groups, aged 18 years and above, were included in the study. The current history of pregnancy and physical deformities were excluded from the study. Clients who were unstable or unable to consent (as determined by the attending physician) were also excluded.

## Sample size determination and sampling technique

The sample size was determined using the analytical study sample size calculation formula by taking a two-sided confidence level of 95%, a power of 80% with a double proportion formula and an equal number of cases to controls [15]. A total sample of 192 study participants (96 psychiatric patients and 96 non-psychiatric individuals) was endorsed after considering a 10% non-response rate [16, 17]. All psychiatric patients were selected using consecutive sampling techniques. After collecting data from a single psychiatric patient, one corresponding age and sex-matched non-psychiatric control was chosen.

## Operational definition

**Body mass Index (BMI)** was calculated as the weight of the individual in kilograms divided by height in meter square. With respect to the body mass index (BMI), there are four groupings: underweight (BMI < 18.5 Kg/m$^2$), normal (BMI between 18.5 and 24.9.Kg/m$^2$), overweight (BMI between 25 and 29.9 Kg/m$^2$), and obese (BMI $\geq$ 30 Kg/m$^2$) [18].

**Vigorous-intensity activity** was defined as any activity that results in a significant increase in breathing or heart rate if performed for at least 75 minutes per week. **Moderate-intensity activity was** defined as any activity that causes a small increase in breathing or heart rate if

continued for at least 150 minutes per week or walking for at least 30 minutes per day. **Sedentary** involves a person not meeting any of the above-mentioned criteria for the moderate- or high-level categories [19].

**Smoking state.** Non-smoker or ever not smoked, which is coded as (0 = no) and all smoker (current, current daily, and past smokers) which is coded as (1 = yes) [20].

**Alcohol consumption.** Ever consumer / consumer of any alcohol represents current alcoholic drinker (past 30 days) which is coded as (0 = No and 1 = Yes) and past alcoholic drinker (drank in the past 12 months) which is coded as (0 = No and 1 = Yes) [21].

### Data collection tools and procedures

After the interview, anthropometric measurements were performed. The height and weight of each subject were measured by using a scale to the nearest 1 cm and 1 kg, respectively. The height of a subject was measured by using an erect height measuring scale. Measurements of height were made with the subject's bare feet. The subjects stood straight against the erect measuring scale, and their heads, shoulders, buttocks, and heels touched the scale. The subject's axis of vision was horizontal. Then, they took a deep breath to relax their shoulders. With a flat object, the upper level of their heads was marked against the scale and measured to the nearest 1 cm. Weight was measured using a weighing scale with light clothing and without shoes.

### Data quality control

The data collection tools prepared in English were translated to the local language and retranslated back to English to confirm the correctness of the translation. Training was given to the data collectors and supervisors about the purpose of the study, measurement technique, and ethical consideration. All questionnaires were checked daily for completeness, accuracy, and clarity by the investigators. A pretest of the data collection tool (questionnaire) was done by pre-testing 5% of the questionnaires at Sabian primary hospital, and necessary corrections were made prior to the actual data collection period.

### Data analysis and report

Data were checked for completeness and entered into Epi-data version 3.1 before being exported to IBM Statistic Package for Social Science (SPSS) Version 21 for analysis. The chi-square test was used to conduct statistical analysis of the differences between groups. The mean scores of continuous variables were compared using an independent sample t-test between groups. Bivariate and multivariate logistic regression with odds ratio were used to determine independent predictors of overweight and obesity.

## Results

### Socio demographic status of study participant

A total of 192 study participants (96 cases and 96 controls) participated in the study to determine the burden of overweight or obesity among severely ill psychiatric patients. The mean age (years) of psychiatric patients was 37.18 ±12.59 and 36.59±13.56 in the non-exposed group, which was a statically insignificant difference (p = 0.754). Around 32.2% (31) of the exposed patients and 44.8% (43) of the non-exposed study participants attended college and above educational level (Table 1).

**Table 1. Sociodemographic characteristics of psychiatric and non-psychiatric controls, Dire Dawa, Eastern Ethiopia 2021.**

| Characteristics | Cases (psychiatric patients) | Controls | *p* value |
|---|---|---|---|
| Sex | | | .561* |
| Male | 55 (57.3%) | 56(58.3%) | |
| Female | 41(42.7%) | 40(41.7%) | |
| Age in years (mean±SD) | 37.18±12.59 | 36.59±13.56 | 0.754** |
| Educational status | | | .076* |
| No formal education | 45(46.9%) | 26(27.1%) | |
| Primary education | 3(-) | 6(6.3%) | |
| Secondary education | 17(17.7%) | 21(21.9%) | |
| College & above | 31(32.2%) | 43(44.8%) | |
| Marital status | | | **0.743*** |
| Single/unmarried | 35(36.5%) | 33(34.4%) | |
| married | 12(12.5%) | 33(34.4%) | |
| Divorced/separated/died | 12(12.5%) | 13(13.5%) | |
| Residence | | | **0.009*** |
| Rural | 59 (61.5%) | 41(42.7%) | |
| Urban | 37(38.5%) | 55 (57.3%) | |
| Employment | | | **0.061*** |
| Unemployed | 72(75.0%) | 36(37.5%) | |
| Employed | 24(25.0%) | 60(62.5%) | |
| Ever smoked | | | 0.86* |
| No | 79 (82.3%) | 82 (85.4%) | |
| Yes | 17 (17.7%) | | |
| Ever Alcohol Intake | | | .401* |
| No | 70(72.9%) | 75 (78.1%) | |
| Yes | 26 (27.1%) | 21 (21.9%) | |
| physical activity | | | **0.521*** |
| Yes | 39(40.6%) | 29(30.2%) | |
| No | 57(59.4%) | 67(69.8%) | |
| Family history of hypertension | | | **0.082*** |
| Yes | 36(37.5%) | 13(13.5%) | |
| No | 69(71.9%) | 83(86.5%) | |
| Family history of diabetes | | | **0.013*** |
| Yes | 27(28.1%) | 13(13.5%) | |
| No | 68(70.8%) | 83(86.5%) | |

*SD = standard deviation*, Independent sample t test,

*Pearson's chi square test.

## The prevalence of overweight/obesity

The prevalence of overweight/obesity was significantly higher among the severely ill psychiatric groups than the in non-exposed controls (p = 0.001). The prevalence of overweight/obesity among psychiatric patients was 43.8% (95% CI: 33.3–55.3%), whereas the magnitude of overweight/obesity among non-psychiatric controls was 20.80% (95% CI: 16.5–28.9%) (Fig 1).

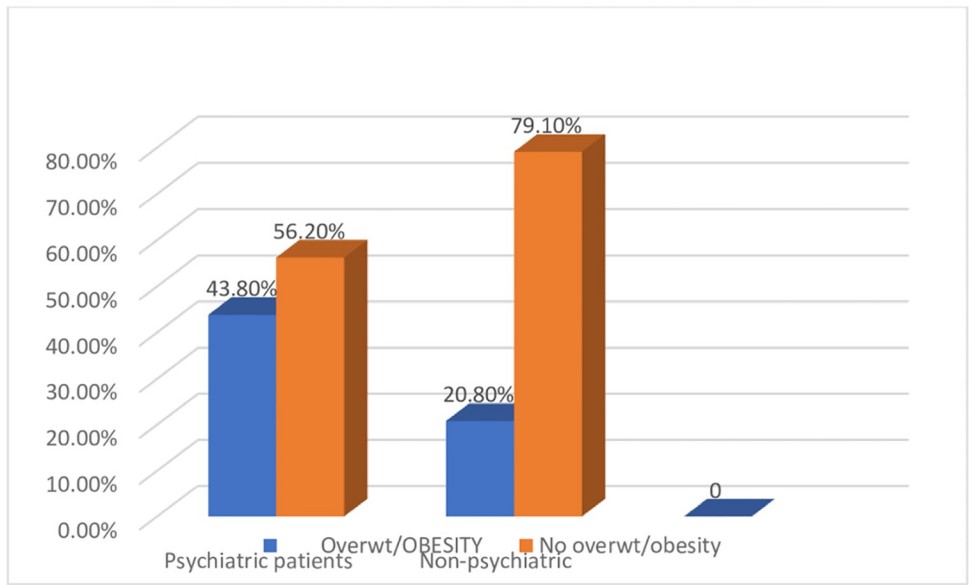

**Fig 1. Shows the prevalence of overweight/obesity among psychiatrically exposed groups and non-psychiatric controls in Dire Dawa, Eastern Ethiopia, 2021.**

## Overweight/obesity patterns in various psychiatric disorders

The prevalence of overweight/obesity was highest in 40% of patients with major depressive disorder (MDD), followed by schizophrenia 32%, bipolar disorder, and 28%. There was a significant correlation between overweight/obesity and the distribution of psychiatric illnesses (Fig 2).

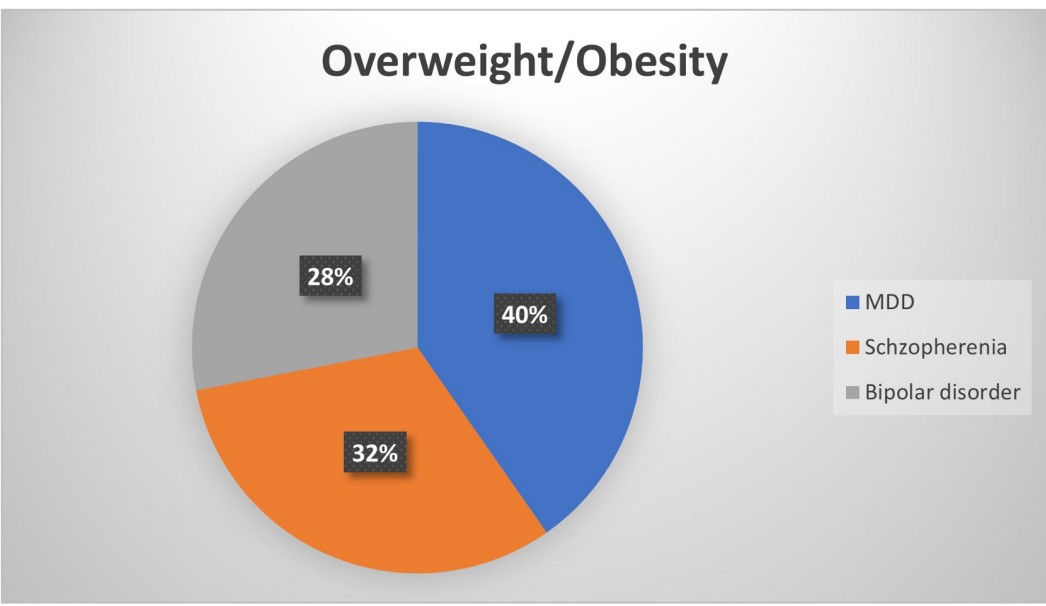

**Fig 2. The Burden of overweight/ obesity among different psychiatric disorders Dire Dawa, Eastern Ethiopia, 2021.**

**Table 2. Factors associated to overweight/ obesity among severely ill psychiatric patients, Dire Dawa, Eastern Ethiopia, 2021.**

| Variables | Category | Overweight/Obesity | | COR (95%CI) | AOR (95%CI) |
|---|---|---|---|---|---|
| | | Yes | No | | |
| Age (years) | 18–30 | 8(26.7%) | 22(73.3%) | 1 | 1 |
| | 31–40 | 12(46.2%) | 14(53.8%) | 2.3(.77,7.20) | 1.4(.33,5.9) |
| | 41–50 | 13(61.9%) | 8(38.1%) | 4.4(1.35,14.7) | 1.6(.3,7.9) |
| | >50 | 9(47.4%) | 10(52.6%) | 2.4(.73,8.3) | .71(.13,3.7) |
| Gender | Male | 20(38.5%) | 32(61.5%) | 1 | 1 |
| | Female | 22(50.0%) | 22(50.0%) | 1.6(.70,3.60) | 1.3(.53,5.02) |
| Marital status | Single | 12(34.3%) | 23(65.7%) | 1 | 1 |
| | Married | 23(46.9%) | 26(53.1%) | 1.6(.69,4.1) | 2.3(.75,7.41) |
| | Widowed | 7(58.3%) | 5(41.7%) | 2.6(.70,10.2) | 2.8(.42,18.8) |
| Education status | Uneducated | 25(55.6%) | 20(44.4%) | 3.6(3.13,18.9) | 2.2(1.57,8.5) |
| | primary | 2(66.7%) | 1(33.3%) | .33(.10,1.1) | 7.7(.16,8.9) |
| | secondary | 5(29.4%) | 12(70.6%) | .38(.14,.99) | 1.1(.19,6.9) |
| | college and above | 10(32.3%) | 21(67.7%) | 1 | 1 |
| Occupation | employed | 3(12.5%) | 21(87.5%) | 1 | 1 |
| | Non employed | 39(54.2%) | 33(45.8%) | 8.2(2.26,30.2) | 6.5(1.8,35.20) |
| Residency | rural | 20(33.9%) | 39(66.1%) | 1 | 1 |
| | urban | 22(59.5%) | 15(40.5%) | 2.8(1.22,6.6) | 1.3(.38,4.5) |
| Smoking | yes | 11(64.7%) | 6(35.3%) | 2.8(.95,8.4) | 2.9(.6,13.5) |
| | No | 31(39.2%) | 48(60.8%) | 1 | 1 |
| Alcohol intake | yes | 13(50.0%) | 13(50.0%) | 1.4(.5,3.4) | 1.01(.25,4.01) |
| | No | 29(41.4%) | 41(58.6%) | 1 | 1 |
| physical exercise | Yes | 14(35.9%) | 25(64.1%) | 1 | 1 |
| | No | 28(49.1%) | 29(50.9%) | 1.7(.74,3.9) | 1.2(.41,3.7) |
| Duration of psychiatric illness (years) | <5 | 10(40.0%) | 15(60.0%) | 1 | 1 |
| | 5–10 | 7(31.8%) | 15(68.2%) | .45(.04,4.2) | 1.3(.28,6.1) |
| | >10 | 25(51.0%) | 24(49.0%) | 3.58(2.10,3.18) | 2.9(1.23,4.1) |

## Factors associated with overweight or obesity

All covariates that with a p-value ≤0.2 in the bivariate analysis were included in the multivariate analysis. In this study age, sex, marital status, educational level, physical activity level, occupational status, and duration of psychiatric illness were significantly associated with overweight or obesity status in the bivariate analysis. After adjusting variables such as study participants who were uneducated (AOR = 2.2, 95% CI (1.57, 8.5), p = 0.014), unemployed (AOR = 6.5, 95% CI (1.8, 35.20), p = 0.023), and duration of psychiatric illness (AOR = 2.9, 95% CI (1.23, 4.1), p = 0.02), were independent predictors of overweight/obesity (Table 2).

## Discussion

To the best of our knowledge, no previous study has been conducted in Ethiopia to determine the prevalence of excess weight among severely ill psychiatric patients. The results showed that the study groups (severely ill psychiatric patients and non-psychiatric controls) were similar in terms of age, sex, marital status, and educational level; with no statistically significant difference.

Our findings revealed a significantly higher prevalence of overweight/obesity (43.8%) among psychiatric patients than among non-psychiatric controls (20.8%), p<0.05. Thus,

possibly due to the deleterious lifestyles characterized by physical inactivity, excessive substance use, and unhealthy diets common among patients with psychiatric disorders, they may be predisposed to overweight or obesity. This study is consistent with the findings reported from the United States, where patients with severe psychiatric disorders had a higher average BMI than the subjects in the general population controls(32.11, SD = 7.72vs 27.62, SD = 5.93, P = 0.000) in 2017 [22], and in Egypt by Ahmed Kamel et al., in 2016 [12]. However, the current study has a higher prevalence of overweight/obesity as compared to the WHO report in 2016 [6] and by Mekonnen et al. in 2018 [23]. This could be due to variations in study participants (in our study, the study participants were severely ill psychiatric patients) and lifestyle and physical activity differences.

Furthermore, the current study shows a higher prevalence of overweight/obesity among those with major depressive disorders (MDD) as compared to schizophrenia and bipolar disorders. This finding is supported by previous studies done in the Campania region by Micanti F, Pecoraro G, Mosca P, et al., 2017 [24] and in McLean Hospital, Belmont, MA, USA by Chouinard, V.-A., et al. [25] and Francesco Weiss, Margherita et al. 2020 [26]. This is partly explained by depressed patients reduced physical activity, which results in positive energy balance and overweight/obesity disorders. Many people who have difficulty recovering from sudden or emotionally draining events unknowingly begin eating too much of the wrong foods or forgoing exercise [27]. Mechanisms underlying this weight gain include lifestyle and environmental factors and psychiatric medications, though emerging evidence has also suggested the role of genetic and neuroendocrine processes [28]. The current study also, found that education status, employment, and increased duration of psychiatric illness were significant predictor of obesity/overweight among psychiatric patients. The current study also found that education status, employment, and increased duration of psychiatric illness were significant predictors of obesity/overweight among psychiatric patients. This finding is consistent with the study conducted by Husky et al. in 2017 [29, 30].

## Limitations of the study

To the best of our knowledge, this is the first study in Ethiopia to provide information on the problem of overweight/obesity among the psychiatric population as compared to the non-psychiatric population. The study's shortcomings include the absence of dietary habits, social desirability, and the possibility that recall biases influenced the results.

## Conclusion

The current study described the burden of overweight or obesity and associated factors among psychiatric patients compared to non-psychiatric controls at Dire Dawa psychiatric center, Ethiopia. The magnitudes of overweight and obesity were significantly higher among the severely ill psychiatric group (43.8%) than in non-exposed controls (20.80%). It was found that educational status, unemployment, and late stages of the disease were significant predictors of overweight or obesity. Clinicians should be aware of the health consequences of overweight and obesity and consider instituting targeted weight treatment programs as a part of routine psychiatric care. This is strongly recommended. The current study also found that education status, non-employment, and the late stage of psychiatric diseases were significant predictors of obesity and overweight. Physicians should be aware of the health consequences of obesity and should consider instituting targeted weight treatment programs as a part of routine psychiatric care.

## Supporting information

**S1 File. Overweight-obesity.**
(DOCX)

**S1 Data. SPSS for overweight.**
(SAV)

## Acknowledgments

The authors forward a special salute to study participants, data collectors, the Eastern Ethiopia mental health center staff, and regional health office managers.

## Author Contributions

**Conceptualization:** Dilnessa Fentie.

**Data curation:** Tariku Derese.

**Formal analysis:** Dilnessa Fentie.

**Investigation:** Dilnessa Fentie, Tariku Derese.

**Methodology:** Dilnessa Fentie.

**Project administration:** Tariku Derese.

**Writing – original draft:** Dilnessa Fentie, Tariku Derese.

**Writing – review & editing:** Dilnessa Fentie, Tariku Derese.

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
