## [Decision Letter · Decision Letter 0]

9 Nov 2021

PONE-D-21-28472Prevalence and Associated Factors of Overweight/Obesity among severely ill Psychiatric Patients in Eastern Ethiopia: A Comparative Cross-Sectional StudyPLOS ONE

Dear Dr. Fentie,

Thank you for submitting your manuscript to PLOS ONE. After careful consideration, we feel that it has merit but does not fully meet PLOS ONE’s publication criteria as it currently stands. Therefore, we invite you to submit a revised version of the manuscript that addresses the points raised during the review process.

We look forward to receiving your revised manuscript.

Kind regards,

Aleksandra Barac

Academic Editor

PLOS ONE

Journal Requirements:

Reviewers' comments:

Reviewer's Responses to Questions

**Comments to the Author**

1. Is the manuscript technically sound, and do the data support the conclusions?

Reviewer #1: Yes

Reviewer #2: Partly

2. Has the statistical analysis been performed appropriately and rigorously? 

Reviewer #1: Yes

Reviewer #2: Yes

3. Have the authors made all data underlying the findings in their manuscript fully available?

Reviewer #1: Yes

Reviewer #2: Yes

4. Is the manuscript presented in an intelligible fashion and written in standard English?

Reviewer #1: No

Reviewer #2: No

5. Review Comments to the Author

Reviewer #1: The overall research hypothesis and how the research question were technically sound and interesting to share with the readers, however the way the data were presented had major problems. Numerous mistakes or inaccuracies with language made some parts of the manuscript difficult to read or follow the idea. Examples include:

- Line 36 : It's ninety six, sex and age matched. Such a mistake in the abstract is very confusing to the reader.

- Line 46: It's weight reduction not treatment, since the scope of your paper is around obesity. The term wight treatment might refer to treating underweight as well.

Other problems with literature review or methodology that had missing important information for example:

- Line 51: I'd rather see statistics from Africa not Europe, since your research is done in Ethiopia. Same data can be obtained from WHO statistics too.

- Line 73: Which sociodemographic characteristics exactly? the was it's written in your review is vague.

- Line 91: There is nothing called non=psychiatric individuals. It could be normal controls not suffering from psychiatric illnesses for example.

- Line 97: Which edition of the DSM? Were the diagnoses made by specialist psychiatrists or GPs and family physicians?

- Line 101: Why were those populations specifically excluded from the study?

I would suggest doing this revisions and also doing an English proof reading by more fluent researcher or possibly whose native language is English and then resubmit.

Reviewer #2: I hope you will find these comments to be helpful

1) some paragraphs need references ; first paragraph of the abstract and introduction ; to make the facts more reliable

2) the authors mentioned that their population was severely ill psychiatric patients, and again they mentioned they were selected from the outpatient! I find this to be contradicting . Do they mean chronic visitors?

3) what are the criteria of severely ill ?

4) Would severely ill patients be able to give a consent ?

5) the manuscript needs language editing

6. PLOS authors have the option to publish the peer review history of their article (what does this mean?). If published, this will include your full peer review and any attached files.

Reviewer #1: No

Reviewer #2: No

---

## [Author Response · Author response to Decision Letter 0]

4 Dec 2021

Comments from Reviewer #1

The overall research hypothesis and how the research question were technically sound and interesting to share with the readers, however the way the data were presented had major problems. Numerous mistakes or inaccuracies with language made some parts of the manuscript difficult to read or follow the idea. 

 The response of the authors: We thank the reviewer for the comment. We have identified the problem and made the necessary corrections throughout the revised manuscript.

Examples include:

1. Line 36 : It's ninety six, sex and age matched. Such a mistake in the abstract is very confusing to the reader.

 The response of the authors: We thank the reviewer for the comment and sorry for vague statements. Correction has made in the revised manuscript (page 2, line no.35-36).

2. Line 46: It's weight reduction not treatment, since the scope of your paper is around obesity. The term wight treatment might refer to treating underweight as well. 

The response of the authors: The reviewer is correct, it is out of our scope and we have modified the contents in the revised manuscript (page 3, line no.48-49)-

Other problems with literature review or methodology that had missing important information for example: 

3. Line 51: I'd rather see statistics from Africa not Europe, since your research is done in Ethiopia. Same data can be obtained from WHO statistics too.

The response of the authors: We thank the reviewer for the suggestions and concern. We have added references indicate about African and Ethiopia to the manuscript on (page 3, line no. 54-56).

4. Line 73: Which sociodemographic characteristics exactly? the was it's written in your review is vague

The response of the authors: We would like to appreciate the reviewer’s comment and concern. We have modified the suggested content to the revised manuscript on (page 4, line no. 87-88).

5. Line 91: There is nothing called non=psychiatric individuals. It could be normal controls not suffering from psychiatric illnesses for example.

The response of the authors: We thank the reviewer for the comment. We have identified the problem and made the necessary corrections in the revised manuscript (page 5, line no.102-105).

6. Line 97: Which edition of the DSM? Were the diagnoses made by specialist psychiatrists or GPs and family physicians?

The response of the authors: The reviewer is correct, and we have incorporated the contents into the revised manuscript (page 5, line no. 100-102). DSM-5 and psychiatrists specialist

7. Line 101: Why were those populations specifically excluded from the study?

Response of the authors: We would like to appreciate the reviewer’s comment and concern. We exclude those who were pregnant and physically deformed study participants because Body Mass Index (BMI) was measured by dividing a person’s weight in kilograms by his or her height in meters squared (kg/m2). The height of the participants was measured using a stadiometer with the participant standing upright with the heel, buttock, and upper back along the same vertical plane. The weight of the participants was measured using a calibrated weighing scale with the participant not wearing shoes and heavy clothes. We believe that physically deformed and pregnant individual measurements might result in abnormal BMI status, thus affecting the final outcomes.

 I would suggest doing this revisions and also doing an English proof reading by more fluent 

 researcher or possibly whose native language is English and then resubmit.

The response of the Authors: We would like to thank you for your comment and we have edited the grammatical issues in the revised manuscript.

Reviewer #2: I hope you will find these comments to be helpful

The response of the Authors: Thank you very much

1) some paragraphs need references ; first paragraph of the abstract and introduction ; to make the facts more reliable

The response of the authors: The reviewer is correct, and we have incorporated the comments into the revised manuscript (page 3, line no.52). However, the first paragraph of the abstract was taken from(references 1,3,4)

2) the authors mentioned that their population was severely ill psychiatric patients, and again they mentioned they were selected from the outpatient! I find this to be contradicting. Do they mean chronic visitors?

The response of the authors: We appreciate the reviewer’s concern and are sorry for the unclarity. Severely ill psychiatric patients (=patients with a diagnosis of one or more of those psychiatric disorders such as schizophrenia, bipolar disorder, major depressive disorder, and schizoaffective disorders) and have chronic follow-up/chronic psychiatric outpatient visitors.

3) what are the criteria of severely ill?

The response of the authors: We would like to appreciate your concern. We were considered the severity of psychiatric disorders as described in DSM-5. Our criteria was based on the number of symptoms, level of distress caused by the intensity of symptoms, and impairment in social and occupational functioning. 

4) Would severely ill patients be able to give a consent?

The response of the authors: We would like to appreciate your concern. We were selected severely psychiatric patients (schizophrenia, bipolar disorder, major depressive disorder, and schizoaffective disorders) and who attended the outpatient psychiatric clinic. Severely ill Clients who were unstable or unable to consent due to acute symptoms (as determined by the attending physician) were excluded. We were considered only stable study participants (those able to give consent).

5) the manuscript needs language editing

The response of the Authors: We would like to thank you for your constructive comments and concerns for forwarding them to us to enrich our paper. We have corrected all the comments and we have also edited the language in the revised manuscript

---

## [Decision Letter · Decision Letter 1]

11 Feb 2022

Prevalence and Associated Factors of Overweight/Obesity among severely ill Psychiatric Patients in Eastern Ethiopia: A Comparative Cross-Sectional Study

PONE-D-21-28472R1

Dear Dr. Fentie,

We’re pleased to inform you that your manuscript has been judged scientifically suitable for publication and will be formally accepted for publication once it meets all outstanding technical requirements.

Kind regards,

Aleksandra Barac

Academic Editor

PLOS ONE

Reviewers' comments:

Reviewer's Responses to Questions

**Comments to the Author**

1. If the authors have adequately addressed your comments raised in a previous round of review and you feel that this manuscript is now acceptable for publication, you may indicate that here to bypass the “Comments to the Author” section, enter your conflict of interest statement in the “Confidential to Editor” section, and submit your "Accept" recommendation.

Reviewer #1: All comments have been addressed

Reviewer #2: All comments have been addressed

2. Is the manuscript technically sound, and do the data support the conclusions?

Reviewer #1: Yes

Reviewer #2: Yes

3. Has the statistical analysis been performed appropriately and rigorously? 

Reviewer #1: Yes

Reviewer #2: Yes

4. Have the authors made all data underlying the findings in their manuscript fully available?

Reviewer #1: Yes

Reviewer #2: Yes

5. Is the manuscript presented in an intelligible fashion and written in standard English?

Reviewer #1: Yes

Reviewer #2: Yes

6. Review Comments to the Author

Reviewer #1: The revision put in consideration all of the comments. Except for 2-3 typing mistakes, the current manuscript is in a much better shape.

Reviewer #2: (No Response)

7. PLOS authors have the option to publish the peer review history of their article (what does this mean?). If published, this will include your full peer review and any attached files.

Reviewer #1: **Yes: **Ahmed Abdelkarim

Reviewer #2: No

---

## [Editor Report · Acceptance letter]

18 Feb 2022

PONE-D-21-28472R1 

Prevalence and Associated Factors of Overweight/Obesity among severely ill Psychiatric Patients in Eastern Ethiopia: A Comparative Cross-Sectional Study 

Dear Dr. Fentie:

I'm pleased to inform you that your manuscript has been deemed suitable for publication in PLOS ONE. Congratulations! Your manuscript is now with our production department. 

Kind regards, 

on behalf of

Dr. Aleksandra Barac 

Academic Editor

PLOS ONE